# Diosgenin Exerts Analgesic Effects by Antagonizing the Selective Inhibition of Transient Receptor Potential Vanilloid 1 in a Mouse Model of Neuropathic Pain

**DOI:** 10.3390/ijms232415854

**Published:** 2022-12-13

**Authors:** Md. Mahbubur Rahman, Hyun Jung Jo, Chul-Kyu Park, Yong Ho Kim

**Affiliations:** Gachon Pain Center and Department of Physiology, College of Medicine, Gachon University, Incheon 21999, Republic of Korea

**Keywords:** diosgenin, transient receptor potential vanilloid 1, neuropathic pain, inflammatory pain, capsaicin

## Abstract

Diosgenin is a botanical steroidal saponin with immunomodulatory, anti-inflammatory, anti-oxidative, anti-thrombotic, anti-apoptotic, anti-depressant, and anti-nociceptive effects. However, the effects of diosgenin on anti-nociception are unclear. Transient receptor potential vanilloid 1 (TRPV1) plays an important role in nociception. Therefore, we investigated whether TRPV1 antagonism mediates the anti-nociceptive effects of diosgenin. In vivo mouse experiments were performed to examine nociception-related behavior, while in vitro experiments were performed to examine calcium currents in dorsal root ganglion (DRG) and Chinese hamster ovary (CHO) cells. The duration of capsaicin-induced licking (pain behavior) was significantly reduced following oral and intraplantar administration of diosgenin, approaching levels observed in mice treated with the TRPV1 antagonist *N*-(4-tertiarybutylphenyl)-4-(3-cholorphyridin-2-yl) tetrahydropyrazine-1(2H)-carbox-amide. Additionally, oral administration of diosgenin blocked capsaicin-induced thermal hyperalgesia. Further, diosgenin reduced capsaicin-induced Ca^2+^ currents in a dose-dependent manner in both DRG and CHO cells. Oral administration of diosgenin also improved thermal and mechanical hyperalgesia in the sciatic nerve constriction injury-induced chronic pain model by reducing the expression of TRPV1 and inflammatory cytokines in DRG cells. Collectively, our results suggest that diosgenin exerts analgesic effects via antagonism of TRPV1 and suppression of inflammation in the DRG in a mouse model of neuropathic pain.

## 1. Introduction

Neuropathic pain (NP) is a common clinical disorder that may arise directly from a lesion or disease and affects the somatosensory system. Various conditions can lead to NP, including trauma to the peripheral or central nervous system (e.g., post-surgical trauma, regional nerve injury, spinal cord injury, and brain injury); neurological diseases (e.g., amyotrophic lateral sclerosis, multiple sclerosis, Guillain–Barré syndrome, syringomyelia); metabolic disorders (Fabry’s disease, diabetic mellitus, alcoholism, and sarcoidosis); and infectious diseases (human immunodeficiency virus infection, coronavirus disease, leprosy, and shingles) [1,2,3,4]. Additionally, NP is a common clinical sign in patients with cancer, resulting from both the disease itself and effects of chemotherapeutic treatment. According to epidemiological data, the prevalence of NP is 6.9–10% [5,6]. Studies have further reported that 25–30% of the population and 60% of adults <65 years of age experience chronic pain [7]. Pain disorders are considered among the most ignored healthcare burdens, despite related expenditures of approximately USD 200 billion annually [7].

Currently, synthetic drugs including opioids, anti-depressants, anti-convulsants, and serotonin–norepinephrine reuptake inhibitors, are used for the treatment of pain disorders [8,9]. However, these drugs have some limitations, including adverse effects such as constipation, cardiovascular side effects, respiratory depression, sedation, and weight gain [10]. Thus, recent pharmacological studies related to pain management have emphasized the need to reduce the use of synthetic drugs [8,10].

Diosgenin is a botanical steroidal saponin present in different plant species (such as *Dioscorea* spp., *Trigonella* spp., *Smilax* spp., *Costus* spp., *Aletris* spp., and *Trillium* spp.) and is especially enriched in fenugreek or wild yam (*Dioscorea* spp.) roots. Studies have demonstrated the various pharmacological effects associated with the use of diosgenin participates, including immunomodulatory, anti-inflammatory, anti-oxidative, anti-thrombotic, anti-apoptotic, anti-depressant, and anti-nociceptive effects [11,12]. Previous studies have demonstrated that diosgenin can reduce pain by up-regulating the Nrf2/HO-1 signaling pathway in diabetic animals, exerting anti-inflammatory [13], and anti-oxidative effects [12,13]. Related studies have also reported that diosgenin reduces pain in an animal model of chronic constriction injury (CCI) by suppressing inflammation, reducing oxidative stress, and restoring anti-oxidative activity [14]. However, the mechanisms by which diosgenin ameliorates pain have not been completely characterized.

The transient receptor potential vanilloid 1 (TRPV1) is non-selective divalent and monovalent cations (e.g., Ca^2+^, Mg^2+^, and Na^+^) channel which is the most important channels involved in the regulation of pain sensitivity. Local sensitization of TRPV1 increases Ca^2+^ influx and release of neuropeptides (e.g., SP, CGRP) leading to neurogenic inflammation as well as triggers changes that lead to oxidative stress, and microglial cell activation, thereby contributing to pain sensation [15]. Interestingly, diosgenin has been shown to significantly block uridine triphosphate-induced Ca^2+^ influx in vascular smooth muscle cells [16], decrease cytoplasmic Ca^2+^ concentrations in human leukemia K562 cells [17], and decrease Ca^2+^ concentrations in cardiac tissue [18]. However, to the best of our knowledge, no studies have investigated the effect of diosgenin on the TRPV1 channel in the context of pain regulation. Therefore, we hypothesized that diosgenin would influence nociceptive behaviors via TRPV1 and investigated the anti-nociceptive effects of diosgenin through nociceptive behavioral studies and in vitro calcium image recordings in dorsal root ganglion (DRG) cells.

## 2. Results

### 2.1. Effects of Diosgenin on Capsaicin-Induced Acute Nociceptive Behavior

Oral treatment with diosgenin at 50 and 100 mg/kg and treatment with the selective TRPV1 antagonist *N*-(4-tertiarybutylphenyl)-4-(3-cholorphyridin-2-yl) tetrahydropyrazine-1(2H)-carbox-amide (BCTC 30 mg/kg) reduced paw-licking time. The paw-licking times for the vehicle + capsaicin 1.6 μg, diosgenin 50 mg/kg, diosgenin 100 mg/kg, and BCTC 30 mg/kg groups were 69.83 ± 7.55 (100%), 54.67 ± 8.48, 42.17 ± 8.18, and 38.67 ± 11.08 s, respectively (Figure 1A), reflecting decreases of 21.72%, 39.62%, and 44.63%, respectively. Direct intraplantar injection of diosgenin also significantly reduced the paw-licking time by 26.13%, 45.56%, and 55.11% in the diosgenin25, diosgenin50, and BCTC0.50 groups, respectively, compared to the NC (100%) (Figure 1B).

Additionally, the paw withdrawal latency (PWL) in the Hargreaves test was significantly lower in the vehicle + capsaicin 1.6 μg group than in the vehicle group (*p* < 0.001) from 30 to 120 min. The PWL was also lower in the diosgenin 50 mg/kg + capsaicin 1.6 μg and BCTC 0.50 μg + capsaicin 1.6 μg groups than that for the vehicle group, although no significant differences were observed until 120 min. However, the PWL was significantly higher in the diosgenin 50 μg + capsaicin 1.6 μg and BCTC 0.50 μg + capsaicin 1.6 μg groups than in the vehicle + capsaicin group from 30 min to 120 min, suggesting the analgesic effects of diosgenin were similar to those of BCTC. The PWL was also reduced in the vehicle group compared to that at baseline (0 min). The two intraplantar vehicle injections also induced pain, but not to the extent observed following capsaicin treatment (Figure 2). Similarly, intraplantar capsaicin injection significantly reduced the paw withdrawal threshold (PWT) in the von Frey test from 30 min to 120 min compared to that in the vehicle group. The PWT was also lower in the diosgenin 50 μg + capsaicin 1.6 μg and BCTC 0.50 μg + capsaicin 1.6 μg groups than in the vehicle group, although no significant differences were observed until 120 min. The PWT was higher in the diosgenin 50 μg + capsaicin 1.6 μg and BCTC 0.50 μg + capsaicin 1.6 μg groups than in the vehicle + capsaicin 1.6 μg group from 30 to 120 min, but significant differences were only observed at 90 and 120 min (Figure 2).

### 2.2. Diosgenin Reduced Transient Capsaicin-Induced Ca^2+^ Currents in Mouse DRG

We first investigated whether diosgenin exhibited any functional interactions with TRPV1 in mouse DRG. Repeated application of capsaicin (100 nM) induced an influx of Ca^2+^ at an extracellular Ca^2+^ concentration of 1 mM (Figure 3A). Following pretreatment with diosgenin for 3 min, capsaicin-evoked intracellular Ca^2+^ transients were significantly reduced by 42.5 ± 10.88% (1 μM, n = 4) and 25.29 ± 5.82% (10 μM, n = 14) compared to the control group (100%) (Figure 3B,C). Therefore, diosgenin could functionally inhibit TRPV1 to reduce Ca^2+^ influx in mouse DRG.

### 2.3. Intracellular Ca^2+^ Concentration in Chinese Hamster Ovary Cells Expressing Human TRPV1

After confirming the normal physiological function of diosgenin in mouse DRG, we evaluated how diosgenin affected TRPV1 using a Chinese hamster ovary (CHO)-K1 cell line expressing human TRPV1. Pretreatment with 1 μM diosgenin for 3 min reduced capsaicin-induced Ca^2+^ influx in human TRPV1 CHO cells (Figure 4A). To investigate the suppressive effect of diosgenin on capsaicin-induced increases in intracellular free calcium ([Ca^2+^]i) in TRPV1-expressing CHO cells, we applied a range of diosgenin concentrations (from 1 nM to 10 μM). Diosgenin inhibited capsaicin-induced activation of human TRPV1 in a dose-dependent manner, with a half-maximal inhibitory concentration of 124.1 nM (Figure 4B). These results indicated that diosgenin directly inhibited TRPV1 activation and that a relatively low concentration of diosgenin could inactivate TRPV1 channels.

### 2.4. Effects of Oral Administration Diosgenin on Rectal Temperature

Rectal temperatures were elevated after oral administration of BCTC and significantly differed only at 60 min (*p* < 0.001) and 90 min (*p* < 0.01) and returned to baseline level after 180 min. Interestingly, there were no significant differences found in both diosgenin treated groups (50 and 100 mg/kg) in the whole experimental period (Figure 5).

### 2.5. Effects of Oral Diosgenin Administration on Body Weight and Nociceptive Behavior

During the experimental period, there were no significant differences in body weight among the study groups. However, on day 10, body weight was lower in the CCI + vehicle control group than in the BCTC 30 mg/kg group. The body weights of the diosgenin-treated groups were higher than those of the BCTC-treated groups and similar to those of the vehicle-treated groups (Figure 6A) indicating the safety of oral diosgenin.

On the baseline day, there were no significant differences in the PWL or PWT between the mouse groups. However, significant decreases in the PWL and PWT were observed in all groups 3 days after CCI + vehicle surgery. Therefore, treatment was initiated on day 3. Our analysis revealed that treatment with diosgenin and BCTC significantly increased the PWL and PWT from day 3 to the end of day 10, while no improvement was observed in the vehicle-treated sham + vehicle group (Figure 6B,C). The PWLs of the sham + vehicle, CCI + Vehicle, CCI + diosgenin 50 mg/kg, CCI + diosgenin 100 mg/kg, and CCI + BCTC 30 mg/kg groups at day 10 were 10.50 ± 0.50, 5.36 ± 0.47, 9.10 ± 0.27, 9.20 ± 0.31, and 9.35 ± 0.20, respectively. The PWTs in these groups were 1.33 ± 0.09, 0.44 ± 0.07, 1.15 ± 0.10, 1.24 ± 0.10, and 1.24 ± 0.13, respectively. The PWL and PWT were significantly higher in both diosgenin (50 and 100 mg/kg) groups and the BCTC 30 mg/kg group than in the CCI + vehicle group. No significant differences were observed between the sham + vehicle, CCI + diosgenin 50 mg/kg, CCI + diosgenin 100 mg/kg, and BCTC 30 mg/kg-treated groups on day 10, suggesting that both 50 mg/kg and 100 mg/kg of diosgenin were therapeutically effective for reducing pain sensitivity in a mouse model of the CCI + vehicle group.

### 2.6. Effects of Diosgenin on Pro-Inflammatory Cytokines and TRPV1 Expression in DRG Tissues

TRPV1 mRNA levels were increased in the CCI + vehicle group, although this effect was significantly attenuated in a dose-dependent manner in the BCTC and diosgenin groups (Figure 7A). In addition, the expression of tumor necrosis factor alpha (TNF-α), interleukin (IL)-1β, and IL-6 was significantly increased in the CCI group compared to that in the sham + vehicle group, a phenomenon that was again attenuated in the BCTC-treated group (Figure 7B–D). Moreover, diosgenin treatment (50 mg/kg and 100 mg/kg) notably repressed the expression of TNF-α, IL-1β, and IL-6 in a dose-dependent manner compared to that in the CCI + vehicle group. These results demonstrated that diosgenin could suppress the TRPV1 and inflammatory response in the DRG after CCI.

## 3. Discussion

Despite several reports related to the analgesic effects of diosgenin in pain models [13,14,19], little is known regarding these effects in the context of nociception. TRPV1 receptors are mainly found in afferent neuronal C fibers and in some Ad fibers. Local sensitization of TRPV1 receptors increases Ca^2+^ influx and aggravates pathological signaling pathways, including neurogenic inflammation in the peripheral nerve, DRG, spinal cord, and brain, resulting in pain sensations [15,20]. Capsaicin is one of the most selective TRPV1 receptor agonists, and local exposure to capsaicin sensitizes the TRPV1 receptor, leading to pain [20]. Therefore, in this study, we investigated whether diosgenin interacts with TRPV1 to ameliorate pain hypersensitivity in mice with capsaicin-induced acute pain behavior. Our findings indicated that oral and intraplantar administration of diosgenin significantly reduced the duration of paw licking induced by intraplantar capsaicin injection, suggesting that diosgenin exerted its effects in part via TRPV1 antagonism. This result was further confirmed by a reduction in capsaicin-induced mechanical and thermal hyperalgesia, as evidenced by the increase in the mechanical PWT and PWL following diosgenin treatment. Consistent with our results, a previous study also reported that diosgenin increased the PWT and PWL in a model of diabetic peripheral NP [19]. Furthermore, we compared the TRPV1-antagonistic effect of diosgenin with that of the standard synthetic TRPV1 antagonist BCTC in capsaicin induced acute pain in mice. As our analysis indicated no significant differences in the duration of paw licking, PWT, or PWL between the diosgenin and BCTC groups, the current results highlight diosgenin as a promising candidate for the development of TRPV1 antagonist drugs.

Capsaicin-evoked intracellular Ca^2+^ transients in DRG [21,22] and CHO cells [23] were measured to evaluate the efficacy of the TRPV1 antagonist candidates. After confirming the antagonistic effects of diosgenin against TRPV1 in a model of capsaicin-induced acute pain, we further analyzed TRPV1 inhibition in an in vitro model of capsaicin-induced TRPV1 activation in mouse DRG contributes to mechanical and heat pain in both mice and rats [24]. Pretreatment with diosgenin significantly inhibited the TRPV1 receptor, as evidenced by the significant reduction in capsaicin-evoked intracellular Ca^2+^ transients in mouse DRG, suggesting TRPV1-antagonistic effects of diosgenin. In addition, for further verification, we measured capsaicin-evoked intracellular Ca^2+^ transients following diosgenin treatment in CHO-K1 cells expressing human TRPV1. These results confirmed that diosgenin treatment directly inhibits TRPV1 activation.

The TRPV1 receptor not only regulates intracellular Na^+^ and Ca^2+^ influx and pain sensation but also serves as a thermosensor for thermoregulation of the body [25]. Most TRPV1 antagonists cause hyperthermia in animal models and humans following systemic intravenous, intraperitoneal, or oral administration [25,26]. However, studies have reported that some TRPV1 antagonists (such as AMG8562, AMG7905, and A-425619) induce hypothermia [25,27]. The TRPV1 antagonist JYL1421 has been shown to induce hyperthermia in monkeys and dogs, but hypothermia in rats [25]. Therefore, thermal alterations are among the most important challenges in the clinical application of TRPV1 antagonists. In this study, we measured changes in rectal temperature following oral administration of diosgenin and BCTC in a time-dependent manner. Oral administration of BCTC significantly increased the rectal temperature in normal mice. Consistently, intravenous administration of BCTC also induced hyperthermia in mice [28]. Although the TRPV1-antagonistic effects of diosgenin were similar to those of BCTC, there were no significant changes in the rectal temperature between the two diosgenin groups.

Many studies have reported that TRPV1 participates in not only acute pain but also persistent pain conditions, especially those related to inflammation, such as the sciatic nerve injury-induced pain model [29,30,31]. After confirmation of TRPV1 antagonism in the capsaicin-induced acute pain model, we evaluated the analgesic effect of diosgenin in a mouse model of CCI-induced chronic pain when compared with that of the TRPV1 antagonist BCTC. Our results indicated that oral administration of diosgenin significantly ameliorated CCI-induced chronic pain, which manifested as an increase in the mechanical PWT and thermal PWL, both of which approached the levels observed in the BCTC group. TRPV1 expression was significantly increased in the CCI-induced pain mice in CCI + vehicle group when compared with that in the sham + vehicle group. These results are consistent with those of previous studies in CCI-induced pain models [29,30] and endometriosis induced pain models [32]. Importantly, oral treatment with diosgenin and BCTC significantly down-regulated TRPV1 expression in the DRG when compared with that in the CCI + vehicle group, further supporting the notion that diosgenin exerts similar effects such as BCTC. It is notable that expression of the TRPV1 down-regulation effect of BCTC in DRG in pain conditions was previously reported [32].

TRPV1 activation mediated Ca^2+^ influx can trigger inflammatory cascades, consequently releasing other pro-inflammatory mediators that contribute to the self-maintenance of chronic inflammation in DRG [33]. Concomitant overexpression of TRPV1 and inflammatory cytokines in the DRG and spinal cord has been reported in chronic pain conditions [29,30,31]. Along with TRPV1, the inflammatory cytokines IL-1β, IL-6, and TNF-α were overexpressed in DRG cells in the CCI group of the present study. Thus, inflammation might contribute to the pathogenesis of neuropathic pain, particularly to the peripheral sensitization. However, diosgenin and BCTC treatment significantly reduced IL-1β, IL-6, and TNF-α expression compared to that observed in the CCI + vehicle group. Thus, diosgenin may have value for reducing the risk of long-term inflammatory complications in patients with chronic inflammatory pain conditions via TRPV1 modulation.

A total of 50 and 100 mg/kg of diosgenin was used orally in this study for nine days. This dosage could be safely manifested by body weights of the diosgenin-treated groups which was similar to those of the vehicle-treated group. Consistently, it was also reported that oral diosgenin administration for 6 weeks has no negative impact on body weight, systemic inflammation, and oxidative stress in normal mice [13], but rather reduced systemic inflammation, oxidative stress and improved body weight and anti-oxidative activities in diabetes animals [13,19]. Oral diosgenin also improved body weight, immune reactivity, and intestinal microbiota in tumor bearing mice [11]. According to previous report, up to 562.5 mg/kg of diosgenin has no harmful effect and 1125 mg/kg or higher dose may result deleterious effect even death [34]. However, diosgenin is a steroidal saponin, intravenous administration may cause hemolysis and may rapidly hydrolyze after oral administration [35]. The limitation of the study is that we did not investigate whether oral diosgenin could induce hemolysis or not. Further study is needed to confirm its safety.

## 4. Materials and Methods

### 4.1. Animals

All experimental protocols were approved by the Institutional Animal Care and Use Committee of the College of Medicine at Gachon University. Male six-week-old C57BL/6 wild-type mice (Orient Bio, Sungnam, Republic of Korea) weighing 18–22 g were used. All experimental animals were housed in a conventional facility under a 12:12-h light/dark cycle (lights on at 8 am) and had ad libitum access to water and chow. Temperature and humidity were maintained at 22 ± 1 °C and 60%, respectively. All animals were acclimatized for 7 days and allowed to habituate to the new environment for 1 week prior to any experiment.

### 4.2. Capsaicin-Induced Acute Nociception and Behavioral Tests

The intraplantar capsaicin test was performed to screen the ability of new TRPV1 antagonists to reduce capsaicin-induced acute nociceptive behavior in experimental animals. To evaluate nociceptive behavior, the time spent licking the paw was recorded in each mouse for 5 min following capsaicin injection. For this purpose, mice were divided into the following four groups (n = 6 each): a capsaicin control group, in which mice were orally administered the vehicle (3% DMSO) only; two diosgenin-treated groups (50 and 100 mg/kg), in which diosgenin was administered at doses of 50 and 100 mg/kg orally, respectively; and a group treated with the TRPV1 antagonist BCTC (BCTC30 mg/kg), in which BCTC was administered 30 mg/kg. A total of 300 µL of vehicle, diosgenin, and BCTC were administered orally 60 min before intraplanar capsaicin injection at 1.6 µg/20 µL per paw [36].

Furthermore, to evaluate the direct effect of diosgenin on pain behavior, mice were divided into another four groups for the intraplantar diosgenin experiments (n = 6 each). In the vehicle + capsaicin 1.6 µg group, capsaicin was injected via the intraplanar route (1.6 µg/20 µL per paw) as previously described [37]. In the two diosgenin-treated groups, 13 µg or 25 µg/10 µL diosgenin was first injected via the intraplantar route. Thirty minutes later, an additional 14 µg/10 µL or 25 µg/10 µL were injected in the diosgenin25 and diosgenin50 groups, respectively. In the BCTC0.50 group, 0.25 µg/10 µL BCTC was injected via the intraplantar route. After 30 min, an additional 0.25 µg were injected. Then, 1.6 µg/10 µL capsaicin was administered concomitantly with the second substance injection. Licking time was also measured as previously mentioned.

Mechanical allodynia was measured using von Frey filaments (NC12775-99; North Coast Medical, CA, USA). The 50% PWT was calculated using the up–down method [38]. Thermal hyperalgesia was measured by recording PWL using a Hargreaves radiant heat apparatus (IITC Life Sciences, Woodland Hills, CA, USA). The cutoff value was set to 20 s to prevent tissue damage. Mechanical allodynia and thermal hyperalgesia were assessed in separate experiments at intervals of 30 min (0, 30, 60, 90, and 120 min). For these experiments, mice were divided into the following four groups (n = 5): capsaicin group, diosgenin50 group, BCTC0.50 group, and NC group injected with vehicle.

### 4.3. Rectal Temperature Recording

The rectal temperature was measured with a digital thermometer (Therma- 1, ETI. Ltd., West Sussex, UK) by inserting a flexible bead probe in to the rectum at the time point of 0, 30, 60, 90, 120, and 240 min after oral administration of 300 µL of vehicle, diosgenin (50 mg/kg and 100 mg/kg), and BCTC 30 mg/kg in normal mice (n = 6).

### 4.4. Preparation of Mouse DRG

DRG tissues were prepared from adult mice (7–9-week-old, male) and incubated with collagenase A (0.2 mg/mL; Roche, Basel, Switzerland)/dispase II (2.4 units/mL; Roche, Basel, Switzerland) at 37 °C for 90 min. The suspension with DRG tissues was triturated with a series of Pasteur pipettes and centrifuged at 3,000 rpm for 3 min. The precipitate was resuspended in neurobasal culture media with 10% FBS, 2% B-27 supplement (Invitrogen, Carlsbad, CA, USA), and 1% penicillin/streptomycin. Cells were plated on a glass coverslip precoated with poly-D-lysine and maintained at 37 °C with 5% CO2 at least 4 h until use.

### 4.5. Intracellular Calcium Imaging

Mechanistically, cell culture human TRPV1 stable CHO-K1 cells (GenBank accession number: NM_080706) were grown in Dulbecco’s modified Eagle’s medium (Welgene, Gyeongsan, Republic of Korea) supplemented with 10% fetal bovine serum (FBS; Gibco, Waltham, MA, USA) and 800 μg/mL Geneticin^™^ Selective Antibiotic (G418 Sulfate; Gibco, Waltham, MA, USA) in a humidified incubator at 37 °C with 5% CO_2_. Passaging of the cells was performed three times per week, reaching a maximum density of 80%.

Cells were allowed to settle in media for 3 min at room temperature (21–25℃), following which fura-2 AM (2 μM; Thermo Fisher Scientific, Waltham, MA, USA) was applied for 40 min at 37 °C. The cells were then rinsed three times with medium and incubated for 30 min. The slides were covered with a cover slip and mounted on an inverted microscope (Olympus BX51WI, Tokyo, Japan) and perfused continuously at 1 mL/min in a bath solution containing 140 Mm NaCl, 5 mM KCl, 1 mM CaCl_2_, 2 mM MgCl_2_, 10 mM HEPES, and 10 mM glucose adjusted to a pH of 7.4 using NaOH. All measurements were performed at room temperature. Cells were illuminated with a 175-W xenon arc lamp, and excitation wavelengths (340/380 nm) were selected using a Lambda DG-4 monochromator filter changer (Shutter Instrument, Novato, CA, USA). The [Ca^2+^]i concentration was measured using digital video microfluorometry with an intensified charge-coupled device camera (OptiMOS, QImaging, Surrey, BC, USA) coupled to a microscope and software (Slidebook 6, 3i, Intelligent Imaging Innovations, Denver, CO, USA).

### 4.6. CCI Model and Behavioral Tests

NP was induced in mice via chronic constriction of the sciatic nerve as previously described [28]. Briefly, mice were anesthetized with 75 mg/kg pentobarbital sodium (Entobar inj. 100 mg, Hanlim Pharm. Co. Ltd., Seoul, Republic of Korea), the right thighs were incised, blunt cuts were made through the biceps, and the sciatic nerve was exposed. Two silk sutures (7–0; Ailee, Busan, Republic of Korea) were loosely knotted around the full circumference of the sciatic nerve 2–3 mm apart. This was carefully performed to prevent disruption of the epineural blood supply. The muscular and skin layers were sutured using a suture silk thread. The sham group underwent surgery on to expose the sciatic nerve, but the nerve was not ligated to cause injury.

Three days after surgery, the animals were divided into the following five groups after confirmation of nociception (n = 10 each): a sham + vehicle group, a sham-operated group orally administered vehicle (3% DMSO); two diosgenin-treated groups (50 and 100 mg/kg), in which oral diosgenin was administered at doses of 50 and 100 mg/kg; and a BCTC 30 mg/kg group (BCTC, 30 mg/kg). In total, 300 µL of vehicle, diosgenin, and BCTC were administered orally daily for 9 days. Both mechanical allodynia and thermal hyperalgesia were induced as described previously at 0, 3, 5, 7, 9, and 10 days before drug administration. These two tests were performed at an interval of 2 h.

### 4.7. Real Time-Quantitative Polymerase Chain Reaction Analysis

Total RNA was isolated from each group of DRG tissues using TRIzol Reagent (Ambion, Austin, TX, USA), in accordance with the manufacturer’s protocol. The yield and purity of isolated RNA were determined using a NanoDrop 2000 spectrophotometer (Thermo Fisher Scientific, Waltham, MA, USA). First-strand cDNA was synthesized from 1 μg of total RNA using M-MLV reverse transcriptase kits (Invitrogen, Carlsbad, CA, USA) in a total volume of 20 μL, in accordance with the manufacturer’s protocol. Real time-quantitative polymerase chain reaction was performed on a QuantStudio 1 system (Thermo Fisher Scientific, Waltham, MA, USA) with a 96-well plate using the SensiFAST SYBR Lo-ROX kit (Meridian Bioscience, Cincinnati, OH, USA) and a reaction volume of 20 μL with 1 μL of sample. The following amplification program was used for all quantitative polymerase chain reactions: 95 °C for 2 min, followed by 50 cycles of 5 s at 95 °C and 30 s at 60 °C. The 2-ΔΔCt values were analyzed using GAPDH as a reference gene. The primers used are listed in Table 1.

### 4.8. Statistical Analyses

All data are expressed as the mean ± standard error of the mean and analyzed by using Prism 5.03 (GraphPad Software Inc., San Diego, CA, USA). The data were statistically analyzed using one-way or two-way RM (repeated measures) analyses of variance (ANOVA) followed by the post hoc Bonferroni test or Dunnett’s test following one-way ANOVA test. The criterion for statistical significance was set at *p* < 0.05.

## 5. Conclusions

Collectively, our results suggest that diosgenin exerts analgesic effects and reduce chronic inflammation via antagonism of TRPV1 in a mouse model of pain, without altering body temperature. The current study provides new insight into the mechanisms underlying the pharmacological effects of diosgenin, which may in turn provide insight into its therapeutic effects. Diosgenin may have value for reducing the risk of long-term inflammatory complications in patients with chronic neuropathic pain via TRPV1 modulation. Therefore, diosgenin represents an effective therapeutic agent in acute and chronic neuropathic pain disorders in mice.

## Figures and Tables

**Figure 1 ijms-23-15854-f001:**
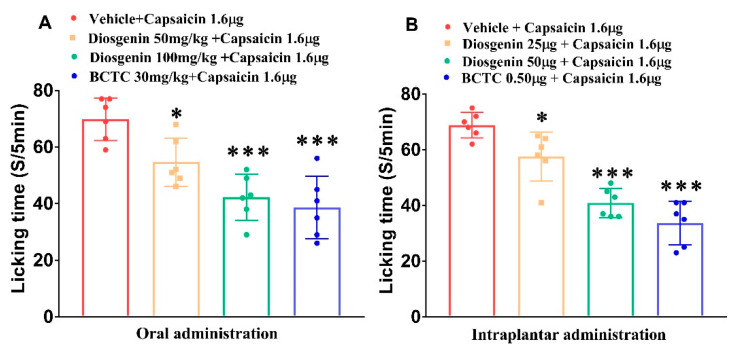
Oral (**A**) and intraplantar (**B**) treatment with diosgenin reduces capsaicin-induced acute licking time. BCTC: *N*-(4-tertiarybutylphenyl)-4-(3-cholorphyridin-2-yl) tetrahydropyrazine-1(2H)-carbox-amide. Data are presented as mean ± standard error of mean (n = 6), * *p* < 0.05 and *** *p* < 0.001, Bonferroni post hoc test following one-way ANOVA versus the vehicle + capsaicin 1.6 μg group.

**Figure 2 ijms-23-15854-f002:**
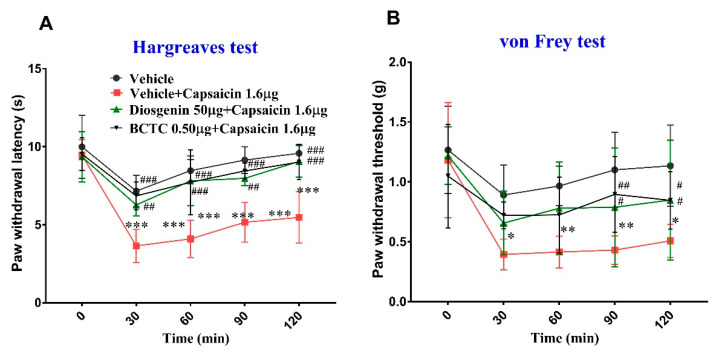
Intraplantar diosgenin treatment reduces capsaicin-induced acute thermal hyperalgesia (**A**) and mechanical allodynia (**B**) in mice; BCTC: *N*-(4-tertiarybutylphenyl)-4-(3-cholorphyridin-2-yl) tetrahydropyrazine-1(2H)-carbox-amide. Data are presented as mean ± standard error of mean (n = 6), * *p* < 0.05, ** *p* < 0.01 and *** *p* < 0.001, Bonferroni post hoc test following two-way RM ANOVA versus the vehicle group; # *p* < 0.05; ## *p* < 0.01; and ### *p* < 0.001, Bonferroni post hoc test following two-way RM ANOVA versus the vehicle + capsaicin 1.6 μg group.

**Figure 3 ijms-23-15854-f003:**
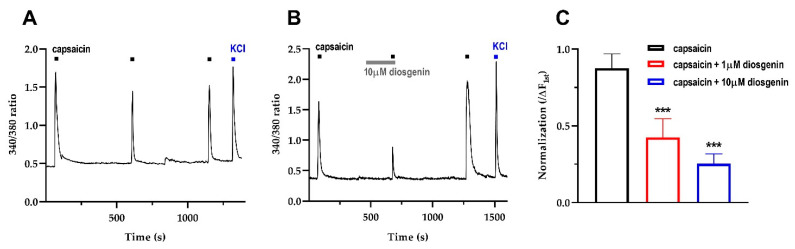
Capsaicin induces an influx of extracellular Ca^2+^ in mouse DRG sensory neurons. (**A**) Capsaicin (100 nM) produces consistent [Ca^2+^]i responses (intervals of 9 to 11 min). (**B**) Pretreatment with 10 μM diosgenin reduces capsaicin-induced [Ca^2+^]i in mouse DRG. (**C**) Mean [Ca^2+^]i responses normalizes to the peak amplitude of the first capsaicin response (n = 6, n = 4, n = 14, respectively). Neuronal cell viability is confirmed based on the response to 50 mM KCl solution (blue, 10 s) at the end of the experiment. Results are presented as mean ± standard error of mean/*** *p* < 0.0001 Dunnett’s test following one-way ANOVA versus the capsaicin group. DRG: dorsal root ganglion; [Ca^2+^]i: intracellular free calcium.

**Figure 4 ijms-23-15854-f004:**
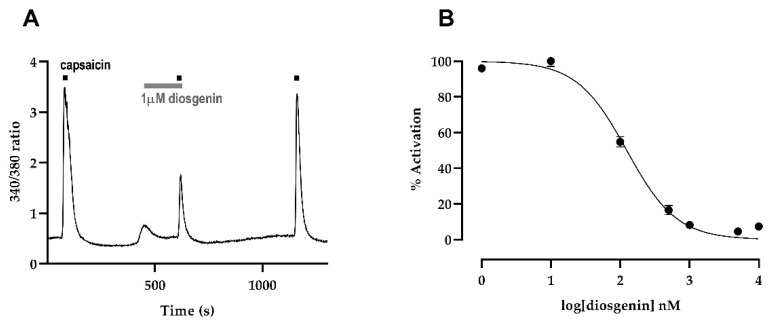
Diosgenin directly inhibits TRPV1 activation in CHO-K1 cells expressing human TRPV1. The capsaicin-induced calcium response is measured in cells pretreated with 1 μM diosgenin (**A**). The half-maximal inhibitory concentration of diosgenin in human TRPV1-expressing CHO-K1 cells. The graph shows that diosgenin antagonizes capsaicin activation in a dose-dependent manner (**B**). Results are presented as mean ± standard error of mean. CHO: Chinese hamster ovary; TRPV1: transient receptor potential vanilloid 1.

**Figure 5 ijms-23-15854-f005:**
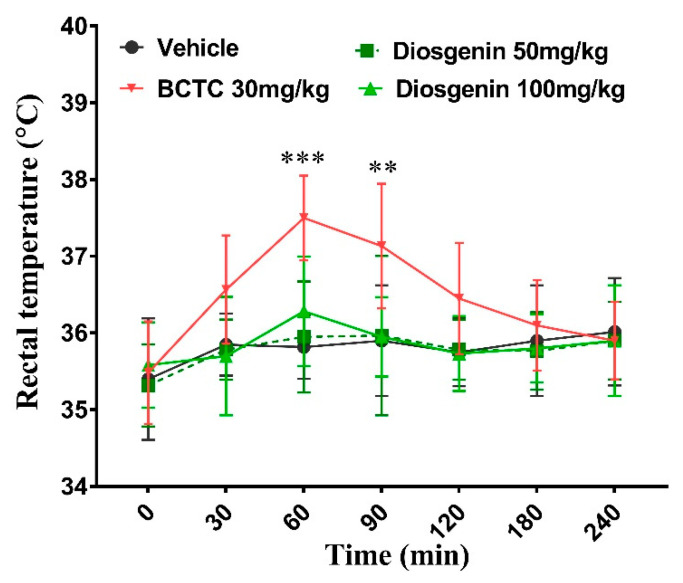
Oral administration diosgenin does not alter rectal temperature in normal mice. BCTC: *N*-(4-tertiarybutylphenyl)-4-(3-cholorphyridin-2-yl) tetrahydropyrazine-1(2H)-carbox-amide. Data are presented as mean ± standard error of mean (n = 6), ** *p* < 0.01 and *** *p* < 0.001, Bonferroni post hoc test following two-way RM ANOVA versus the vehicle group.

**Figure 6 ijms-23-15854-f006:**
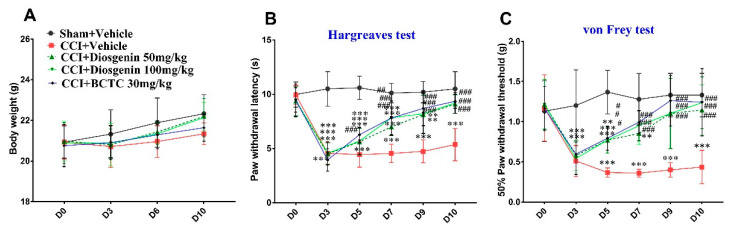
Effects of diosgenin on body weight (**A**), paw withdrawal latency (**B**), and paw withdrawal threshold (**C**) in a mouse model of chronic constriction injury.; BCTC: *N*-(4-tertiarybutylphenyl)-4-(3-cholorphyridin-2-yl) tetrahydropyrazine-1(2H)-carbox-amide; CCI: chronic constriction injury. Data are presented as mean ± standard error of mean (n = 10), ** *p* < 0.05 and *** *p* < 0.001, Bonferroni post hoc test following two-way RM ANOVA versus the sham + vehicle group; # *p* < 0.05; ## *p* < 0.01; and ### *p* < 0.001, Bonferroni post hoc test following two-way RM ANOVA versus the CCI + vehicle group.

**Figure 7 ijms-23-15854-f007:**
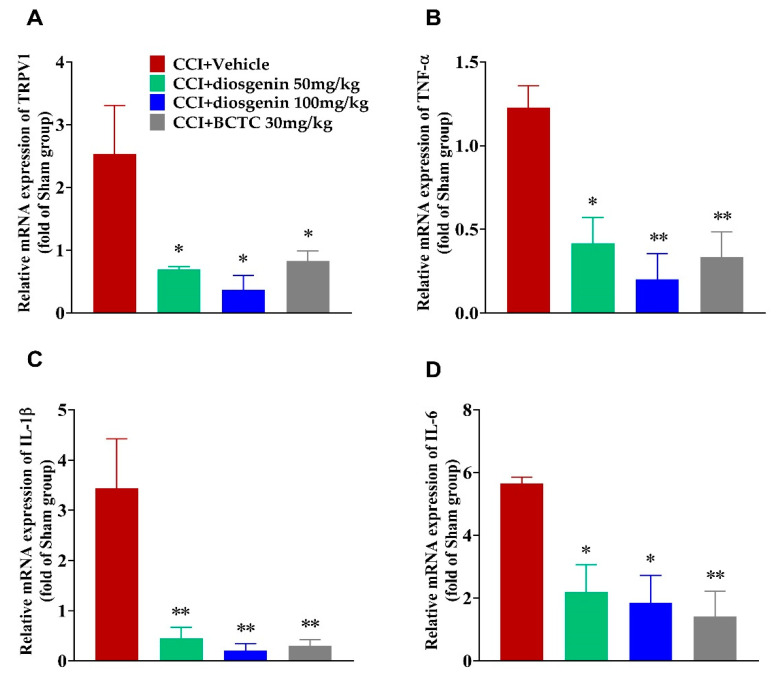
Relative quantification of (**A**) TRPV1, (**B**) TNF-α, (**C**) IL-1β, and (**D**) IL-6 mRNA levels in the DRG of different groups, determined using the 2^-ΔΔCq^ method following normalization to internal controls. Data are presented as mean ± standard error of mean (all n = 3). * *p* < 0.05 and ** *p* < 0.001, Dunnett’s test following one-way ANOVA versus the CCI + vehicle group. CCI: chronic constriction injury; TRPV1: transient receptor potential vanilloid 1; TNF-α: tumor necrosis factor alpha; IL: interleukin.

**Table 1 ijms-23-15854-t001:** Primers used for RT-qPCR amplification.

Target GENE (Product Length)	Forward	Reverse
Mouse TRPV1 (203 bp)	AAGGCTTGCCCCCCTATAA	CACCAGCATGAACAGTGACTGT
Mouse TNF-α (206 bp)	CGTCAGCCGATTTGCTATCT	CGGACTCCGCAAAGTCTAAG
Mouse IL-1β (140 bp)	TGACGGACCCCAAAAGATGA	TCTCCACAGCCACAATGAGT
Mouse IL-6 (228 bp)	AGACTTCCATCCAGTTGCCT	AGTGCATCATCGTTGTTCATACA
Mouse GAPDH (150 bp)	TGTGTCCGTCGTGGATCTGA	TTGCTGTTGAAGTCGCAGGAG

TRPV1: transient receptor potential vanilloid 1; TNF-α: tumor necrosis factor alpha; IL: interleukin; GAPDH: glyceraldehyde-3-phosphate dehydrogenase.

## Data Availability

All data are stored at Gachon Pain Center and Department of Physiology, College of Medicine, Gachon University, Incheon 21999, Republic of Korea and data will be available upon request following appropriate region.

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
