# Peer review of "Diosgenin Exerts Analgesic Effects by Antagonizing the Selective Inhibition of Transient Receptor Potential Vanilloid 1 in a Mouse Model of Neuropathic Pain"

_ijms, 2022, doi:10.3390/ijms232415854_

Round 1

Reviewer 1 Report

Results are set before the Materials and Methods!

Please change the order of the headings because it makes a mess of the readability...

Line 161  - baseline …

Line 242- thermosensor…

Page 9, line 321 - The data concerning cell cultures is inserted under the 4.3 heading, Rectal temperature reading (?)

The discussions are all over the place... please try to itemize the results, use a little less acronyms, because the readability is poor and the conclusions difficult to justify

The paper is very good, but difficult to read and to follow...

Author Response

Open Review

English language and style

( ) English very difficult to understand/incomprehensible
( ) Extensive editing of English language and style required
( ) Moderate English changes required
(x) English language and style are fine/minor spell check required
( ) I don't feel qualified to judge about the English language and style

Yes

Can be improved

Must be improved

Not applicable

Does the introduction provide sufficient background and include all relevant references?

(x)

( )

( )

( )

Are all the cited references relevant to the research?

(x)

( )

( )

( )

Is the research design appropriate?

(x)

( )

( )

( )

Are the methods adequately described?

( )

(x)

( )

( )

Are the results clearly presented?

(x)

( )

( )

( )

Are the conclusions supported by the results?

( )

(x)

( )

( )

Comments and Suggestions for Authors

  1. Results are set before the Materials and Methods!

Please change the order of the headings because it makes a mess of the readability...

Reply: Dear reviewer, your absolutely right. But this is the journal style. We just followed the journal’s instruction. https://www.mdpi.com/1422-0067/23/21/13662 . Could you please see the recently published article.

  1. Line 161  - baseline …

Reply: corrected

“Rectal temperatures were elevated after oral administration of BCTC and significantly differed only at 60 min (p < 0.001) and 90 min (p < 0.01) and returned to baseline level after 180 min”.

  1. Line 242- thermosensor…

Reply: corrected

“The TRPV1 receptor not only regulates intracellular Ca2+ influx and pain sensation but also serves as a thermosensor for thermoregulation of the body [24].

  1. Page 9, line 321 - The data concerning cell cultures is inserted under the 4.3 heading, Rectal temperature reading (?)

Reply: It was great mistake. We have transferred the these sentences to the “ 4.5. Intracellular calcium imaging” from “ 4.3. Rectal temperature recording”

4.3. Rectal temperature recording

The rectal temperature was measured by inserting a flexible bead probe with a digital thermometer (Therma- 1, ETI. Ltd, West Sussex, UK) at 0, 30, 60, 90, 120, and 240 min after oral administration of 300 µL of vehicle, diosgenin (50 mg/kg and 100 mg/kg), and BCTC 30 mg/kg in normal mice (n = 6).

4.5. Intracellular calcium imaging

Mechanistically, cell culture human TRPV1 stable CHO-K1 cells (GenBank accession number: NM_080706) were grown in Dulbecco’s modified Eagle’s medium (Welgene, Gyeongsan, Korea) supplemented with 10 % fetal bovine serum (FBS; Gibco, Waltham, MA, USA) and 800 μg/mL Geneticin Selective Antibiotic (G418 Sulfate; Gibco, Waltham, MA, USA) in a humidified incubator at 37°C with 5% CO2. Passaging of the cells was performed three times per week, reaching a maximum density of 80%.

Cells were allowed to settle in media for 3 min at room temperature (21-25℃), following which fura-2 AM (2 μM; Thermo Fisher Scientific, Waltham, MA, USA) was applied for 40 min at 37°C. The cells were then rinsed three times with medium and incubated for 30 min. The slides were covered with a cover slip and mounted on an inverted microscope..

  1. The discussions are all over the place... please try to itemize the results, use a little less acronyms, because the readability is poor and the conclusions difficult to justify

Reply: We have changed the full manuscript including figures by decreasing the acronyms after consideration of your valuable suggestion.

  • DG to diosgenin
  • CPS to capsaicin

  1. The paper is very good, but difficult to read and to follow...

Reply: You are appreciated for this nice comment. We have reduced acronyms. Hope it would be easy to read and follow.

Submission Date

10 November 2022

Date of this review

17 Nov 2022 10:18:41

Reviewer 2 Report

First of all, the manuscript is well-written with high quality of language, and describes a well-designed and well-conducted study. However, I would have some formal and professional comments, which can improve the manuscript.

Abstract:

Line 14: The authors wrote in Line 13 that DG has anti-nociceptive effects, but in Line 14 they stated that its effects on nociception are unclear. Please rephrase this sentence, I think the authors thought about the unclear mechanisms of anti-nociceptive effects of DG.

Line 25: CCI model is a well-known and widely used NEUROPATHIC, not a chronic inflammatory pain model, as the authors refers to it in the title of the manuscript. Of course we know that low-grade inflammation and some inflammatory mechanisms can contribute to the pathogenesis of neuropathic pain, particularly to the peripheral sensitization, but it is a strong statement that it would be a chronic inflammatory pain model. This criticism also applies to the last paragraph of the discussion section (Line 256, 259 and 277). Please correct this statement.

Line 28: Since the authors used two types of pain models during the study, and in the last sentence they refer only to the CCI model, they should specify the name of the model there. I recommend the use of “in a mouse model of neuropathic pain”.

Introduction:

Line 35: Please put “or” before the “central nervous system”.

Line 38: As I know, CRPS is no longer involved in the etiology of neuropathic pain. Based on the latest recommendation and classification of IASP it is characterized by nociplastic pain.

Line 47-52: The authors specified neuropathic pain in the title and at the beginning of the introduction section, therefore it is strange that the authors talk about the treatment of pain disorders generally. Of course the treatments are overlapped, but in this case they should better focus on the treatment of NP, in which for example NSAIDs are ineffective.

Line 66-68: The authors oversimplified the function of TRPV1. It is a ligand-gated ion channel (ionotropic receptor), which is permeable for both Na+ and Ca2+ ions. Na+ influx is responsible for the development of action potential and pain sensation, while Ca2+ influx for the release of neuropeptides (e.g. SP, CGRP etc.) leading to neurogenic inflammation. Please revise this sentence. This criticism also applies to the first paragraph of the discussion section (Line 212-215).

Line 69: Please specify what “UTP does mean.

Results:

General comments:

I recommend that the authors should correct the name of the NC group in the whole manuscript, also on the figures. The name of Vehicle-treated group (abbreviated Vehicle or VEH) would be much appropriate.

Please specify in all figure legends that Mean ± S.E.M. was demonstrated and put “versus” after the p values (e.g. *p<0.05 versus CPS+VEH group).

Figure 1.:

-          Please standardize the name of the legends, CPS should be placed for the first place in the case of the control group as well similarly to the legends of the treated groups.

-          Panel A: In the legends of the treated groups the authors should specify the doses (e.g. CPS+DG100mg/kg, only 100 mg is not enough).

-          Line 94: The use of two-way ANOVA is not appropriate here, because the authors have only one variable (treatment), one-way ANOVA should be much better.

Line 97: What does “C group” mean?

Line 97 and 99: The names of the treated groups are not equal with their names presented on the figures.

Figure 2.:

-          Please represent on the figure, if NC and CPS 1.6ug groups got Vehicle (NC+Veh, CPS 1.6ug+Veh). Instead of NC I recommend Sham.

-          Line 111 and 112: The panels are swapped.

-          Line 114: The authors should use two-way RM ANOVA instead of ordinary two-way ANOVA.

-          Line 115: versus CPS instead of versus CCI

Figure 3: Statistics is missing form the legend.

Figure 5.:

-          Please simplify the name of the legends. NC is redundant, I recommend legends of Vehicle, DG50mg/kg, DG100mg/kg and BCTC30mg/kg.

-          Line 156: The authors should use two-way RM ANOVA instead of ordinary two-way ANOVA.

Line 165: Please specify the “nociceptive behavior” (CCI-induced).

Figure 6.:

-          Please represent on the figure, if NC and CCI groups got Vehicle (NC+Veh, CCI+Veh). Instead of NC I recommend Sham.

-          Line 188 and 190: The authors should use two-way RM ANOVA instead of ordinary two-way ANOVA.

Line 191-192: “DRG tissues of CCI model mice”? Please rephrase the tile of the subheading.

Line 200: Based on which do the authors state that the inflammatory responses are TRPV1-related?

Line 207: Statistics is missing.

Discussion:

Line 219: Which factors were the oral use of DG allowed? Is it reliably absorbed from the gut? Are other saponins also orally used? What do we know about its toxicity and tolerability? Do you have data about is? If yes, please refer to it briefly.

Line 226: The authors wrote here about CPS model, not about CCI model.

Line 230-231: “were measured to evaluate the efficacy of the TRPV1 agonist and antagonist candidates”? Please rephrase the sentence.

Line 234-235: contributes to mechanical and HEAT pain in both mice and humans

Line 242: thermosensor

Line 251: What does “normal mice” mean? Shame mice?

Line 259: TRPV1 antagonist, not agonist

Line 262 and 268: Why does TRPV1 expression decrease confirm the TRPV1 antagonistic effect of DG? It is not necessary that an antagonist decreases the expression of its target receptor. The use of an antagonist can better upregulate its target receptor as a compensatory effect, if it is continuously used. What would be the mechanism of TRPV1 expression decrease? Please revise this statement.

Line 270: Please specify from which cells TRPV1 activation can release pro-inflammatory mediators.

Line 272: patients with chronic NEUROPATHIC pain

Materials and Methods:

Line 282: Age of the animals is missing.

Line 295: Please define which vehicle was used.

Line 298 and 363: How (by an oral gavage) and in which volume (? ml/10 g?) were the substances used? Please define.

Line 304, 307 and 365: Please specify that CAPS (1.6 ug/10 ul) was administered concomitantly with the second substance injection.

Line 317: The details of rectal temperature recording are missing. Please describe detailed the method.

Line 321: Sham mice instead of normal mice?

Line 321-327: This part belongs to another method description.

Line 386: Which software was used to the statistical analysis?

Line 386-390: Please revise this section…

Line 392: Regulates chronic inflammation? Please revise this statement.

Line 396: I have never heard about inflammatory neuropathic pain… Please remove the indicator of inflammatory…

Author Response

Open Review

English language and style

( ) English very difficult to understand/incomprehensible
( ) Extensive editing of English language and style required
( ) Moderate English changes required
(x) English language and style are fine/minor spell check required
( ) I don't feel qualified to judge about the English language and style

Yes

Can be improved

Must be improved

Not applicable

Does the introduction provide sufficient background and include all relevant references?

( )

( )

(x)

( )

Are all the cited references relevant to the research?

(x)

( )

( )

( )

Is the research design appropriate?

(x)

( )

( )

( )

Are the methods adequately described?

( )

( )

(x)

( )

Are the results clearly presented?

( )

( )

(x)

( )

Are the conclusions supported by the results?

( )

( )

(x)

( )

Comments and Suggestions for Authors

First of all, the manuscript is well-written with high quality of language, and describes a well-designed and well-conducted study. However, I would have some formal and professional comments, which can improve the manuscript.

Abstract:

  1. Line 14: The authors wrote in Line 13 that DG has anti-nociceptive effects, but in Line 14 they stated that its effects on nociception are unclear. Please rephrase this sentence, I think the authors thought about the unclear mechanisms of anti-nociceptive effects of DG.

Reply: Corrected the mistake

“Diosgenin (DG) is a botanical steroidal saponin with immunomodulatory, anti-inflammatory, anti-oxidative, anti-thrombotic, anti-apoptotic, anti-depressant, and anti-nociceptive effects. However, the effects of DG on anti-nociception are unclear”.

  1. Line 25: CCI model is a well-known and widely used NEUROPATHIC, not a chronic inflammatory pain model, as the authors refers to it in the title of the manuscript. Of course we know that low-grade inflammation and some inflammatory mechanisms can contribute to the pathogenesis of neuropathic pain, particularly to the peripheral sensitization, but it is a strong statement that it would be a chronic inflammatory pain model. This criticism also applies to the last paragraph of the discussion section (Line 256, 259 and 277). Please correct this statement.

Reply: We have modified by deleting “inflammatory” word after considering the valuable comments from Line 25, 256-259.

Abstract:

Line 25: Oral administration of diosgenin also improved thermal and mechanical hyperalgesia in the sciatic nerve constriction injury-induced chronic pain model by reducing the expression of TRPV1 and inflammatory cytokines in DRG cells.

Line 256, 259: After confirmation of TRPV1 antagonism in the capsaicin-induced acute pain model, we evaluated the analgesic effect of diosgenin in a mouse model of CCI-induced chronic pain when compared with that of the TRPV1 agonist BCTC.

Line 277: We think in line 277, “inflammatory” word matched because it was just after the inflammatory statement. Furthermore, we have added your valuable comment in our manuscript, hope it will ornament the statement of inflammation.

“Along with TRPV1, the inflammatory cytokines IL-1β, IL-6, and TNF-α were overexpressed in DRG cells in the CCI group of the present study. Thus, inflammation contributes to the pathogenesis of neuropathic pain, particularly to the peripheral sensitization. However, diosgenin and BCTC treatment significantly reduced IL-1β, IL-6, and TNF-α expression compared to that observed in the CCI group. Thus, diosgenin may have value for reducing the risk of long-term inflammatory complications in patients with chronic inflammatory pain conditions via TRPV1 modulation”.

  1. Line 28: Since the authors used two types of pain models during the study, and in the last sentence they refer only to the CCI model, they should specify the name of the model there. I recommend the use of “in a mouse model of neuropathic pain”.

Reply: Modified according to your suggestion

“Taken together, our results suggest that diosgenin exerts analgesic effects via antagonism of TRPV1 and suppression of inflammation in the DRG in a mouse model of neuropathic pain”.

Introduction:

  1. Line 35: Please put “or” before the “central nervous system”.

Reply: Corrected the mistake

 “including trauma to the peripheral, or central nervous system”

  1. Line 38: As I know, CRPS is no longer involved in the etiology of neuropathic pain. Based on the latest recommendation and classification of IASP it is characterized by nociplastic pain.

Reply: Deleted

“neurological diseases (e.g., amyotrophic lateral sclerosis, multiple sclerosis, Guillain–Barré syndrome, syringomyelia, etc.)”;

  1. Line 47-52: The authors specified neuropathic pain in the title and at the beginning of the introduction section, therefore it is strange that the authors talk about the treatment of pain disorders generally. Of course the treatments are overlapped, but in this case they should better focus on the treatment of NP, in which for example NSAIDs are ineffective.

Reply: NSAID Deleted and focused on neuropathic pain

“Currently, synthetic drugs including  opioids, anti-depressants, anti-convulsants, and serotonin–norepinephrine reuptake inhibitors, are used for the treatment of pain disorders [8,9].”.

  1. Line 66-68: The authors oversimplified the function of TRPV1. It is a ligand-gated ion channel (ionotropic receptor), which is permeable for both Na+ and Ca2+ ions. Na+ influx is responsible for the development of action potential and pain sensation, while Ca2+ influx for the release of neuropeptides (e.g. SP, CGRP etc.) leading to neurogenic inflammation. Please revise this sentence. This criticism also applies to the first paragraph of the discussion section (Line 212-215).

Reply: Rewrote the sentences according to your suggestion-

Line 66-68:

Na+ and Ca2+ influx through the TRPV1 channel triggers changes that lead to oxidative stress, inflammatory cascades, and microglial cell activation, thereby contributing to pain sensation [14]”.

Line 212-215

“Local sensitization of TRPV1 receptors increases Na+ and Ca2+ influx and aggravates pathological signaling pathways, including inflammatory cascades, which induce nerve depolarization transmitted to the spinal cord and brain, resulting in pain sensations [14,19].

  1. Line 69: Please specify what “UTP does mean.

Reply: Corrected the mistake

“Interestingly, DG has been shown to significantly block uridine triphosphate-induced Ca2+ influx in vascular smooth muscle cells [15]”.

Results:

General comments:

  1. I recommend that the authors should correct the name of the NC group in the whole manuscript, also on the figures. The name of Vehicle-treated group (abbreviated Vehicle or VEH) would be much appropriate.

 Reply: NC group has been changed to Vehicle group in the whole manuscript including figures according to your suggestion.

  1. Please specify in all figure legends that Mean ±S.E.M. was demonstrated and put “versus” after the p values (e.g. *p<0.05 versus CPS+VEH group).

Reply: Corrected

“Oral and intraplantar treatment with diosgenin reduces capsaicin-induced acute licking time. BCTC: N-(4-tertiarybutylphenyl)-4-(3-cholorphyridin-2-yl) tetrahydropyrazine-1(2H)-carbox-amide. Data are presented as mean ± standard error of mean (n = 6), *p < 0.05 and ***p < 0.001, Bonferroni post hoc test following one-way ANOVA versus the Vehicle+capsaicin1.6ug group”.

  1. Figure 1.:

-  Please standardize the name of the legends, CPS should be placed for the first place in the case of the control group as well similarly to the legends of the treated groups.

Reply: We have changed these all groups name.

Oral dose: First we administered vehicle, diosgenin or BCTC, after one hour intraplantar injection of capsaicin.

Intraplantar injection: Actually, we administered vehicle or therapeutic agent by two injections; First injection half dose of diosgenin or BCTC, after 30 min second injection rest half dose of diosgenin or BCTC mixing with capsaicin.  We clearly described in the “Materials and method” section.

As vehicle or therapeutic agent first administered, then Capsaicin, We changed the group name –

For Licking time:

Oral administered groups:

Vehicle+Capsaicin1.6μg,

Diosgenin50mg/kg+Capsaicin1.6μg

Diosgenin50mg/kg+Capsaicin1.6μg

BCTC30mg/kg+Capsaicin1.6μg

Intraplantar administerd groups:

Vehicle+Capsaicin1.6μg,

Diosgenin25μg +Capsaicin1.6μg

Diosgenin50μg +Capsaicin1.6μg

BCTC0.50μg +Capsaicin1.6μg

For mechanical and thermal hyperalgesia:

Vehicle

Vehicle+Capsaicin1.6μg,

Diosgenin50μg +Capsaicin1.6μg

BCTC0.50μg +Capsaicin1.6μg

  • Panel A: In the legends of the treated groups the authors should specify the doses (e.g. CPS+DG100mg/kg, only 100 mg is not enough).

Reply: We have changed these all groups name.

Oral administered groups for Licking time:

Vehicle+Capsaicin1.6μg,

Diosgenin50mg/kg+Capsaicin1.6μg

Diosgenin50mg/kg+Capsaicin1.6μg

BCTC30mg/kg+Capsaicin1.6μg

For mechanical and thermal hyperalgesia in CCI mouse:

We have changed these all groups name. Here first CCI was performed, after confirmation of pain sensitivity, vehicle, diosgenin or BCTC were administered until 9 days. So in the group name CCI was first used. mg/kg added in all groups. The groups are-

Sham+Vehicle

CCI+Diosgenin50mg/kg

CCI+Diosgenin100mg/kg

CCI+BCTC30mg/kg

  • Line 94: The use of two-way ANOVA is not appropriate here, because the authors have only one variable (treatment), one-way ANOVA should be much better.

Reply: Corrected the mistake

Figure 1. Oral and intraplantar treatment with diosgenin reduces capsaicin-induced acute licking time. BCTC: N-(4-tertiarybutylphenyl)-4-(3-cholorphyridin-2-yl) tetrahydropyrazine-1(2H)-carbox-amide. Data are presented as mean ± standard error of mean (n = 6), *p < 0.05 and ***p < 0.001, Bonferroni post hoc test following one-way ANOVA versus the Vehicle+ capsaicin1.6ug group.

”.

  • Line 97: What does “C group” mean?

Reply: Corrected the mistake, C group to vehicle group

“The PWL was also lower in the Diosgenin 50mg/kg+Capsaicin 1.6μg and BCTC 0.50μg+Capsaicin 1.6μg groups than that for Vehicle group, although no significant differences were observed until 120 min.”.

  • Line 97 and 99: The names of the treated groups are not equal with their names presented on the figures.

 Reply: Corrected the mistake

“However, the PWL was significantly higher in the diosgenin 50μg+capsaicin 1.6μg and BCTC 0.50μg+capsaicin 1.6μg groups than in the capsaicin group from 30 min to 120 min, suggesting the analgesic effects of diosgenin were similar to those of BCTC. The PWL was also reduced in the vehicle group compared to that at baseline (0 min)”.

  1. Figure 2.:

-          Please represent on the figure, if NC and CPS 1.6ug groups got Vehicle (NC+Veh, CPS 1.6ug+Veh). Instead of NC I recommend Sham.

Reply: Group name has been changed

For mechanical and thermal hyperalgesia:

Vehicle

Vehicle+Capsaicin 1.6μg,

Diosgenin 50μg +Capsaicin 1.6μg

BCTC 0.50μg +Capsaicin 1.6μg

-          Line 111 and 112: The panels are swapped.

Reply: Corrected the mistake

Figure 2. Intraplantar diosgenin treatment reduces capsaicin-induced acute thermal hyperalgesia (A) and mechanical allodynia (B) in mice.

-          Line 114: The authors should use two-way RM ANOVA instead of ordinary two-way ANOVA.

Reply: Dear reviewer we analyzed by using Prism 5.03 (GraphPad Software Inc., San Diego, CA, USA)” and two-way ANOVA analysis is nicely matched.

-          Line 115: versus CPS instead of versus CCI

 Reply: Corrected the mistake

Figure 2. Intraplantar diosgenin treatment reduces capsaicin-induced acute thermal hyperalgesia (A) and mechanical allodynia (B) in mice. ; BCTC: N-(4-tertiarybutylphenyl)-4-(3-cholorphyridin-2-yl) tetrahydropyrazine-1(2H)-carbox-amide. Data are presented as mean ± standard error of mean (n = 6), *p < 0.05, *p < 0.01 and ***p < 0.001, Bonferroni post hoc test following two-way ANOVA versus the Vehicle group; #p < 0.05; ##p < 0.01; and ###p < 0.001, Bonferroni post hoc test following two-way ANOVA versus the Vehicle+Capsaicin1.6μg group.

  1. Figure 3: Statistics is missing from the legend.

Reply: Statistics are added in Figure 3

“Results are presented as mean ± standard error of mean (***p < 0.0001) Dunnett's test following one-way ANOVA versus the capsaicin group. DRG: dorsal root ganglion; [Ca2+]i: intracellular free calcium”.

  1. Figure 5.:

-          Please simplify the name of the legends. NC is redundant, I recommend legends of Vehicle, DG50mg/kg, DG100mg/kg and BCTC30mg/kg.

Reply: Group name changed for temperature recording

-          Line 156: The authors should use two-way RM ANOVA instead of ordinary two-way ANOVA.

Reply: Dear reviewer we analyzed by using Prism 5.03 (GraphPad Software Inc., San Diego, CA, USA)” and two-way ANOVA analysis is nicely matched.

  1. Line 165: Please specify the “nociceptive behavior” (CCI-induced).

Reply: In the first paragraph, we have described about the body weight of mice.

In the second paragraph, we have described about the nociceptor behavior  

“2.5. Effects of oral diosgenin administration on body weight and nociceptive behavior

During the experimental period, there were no significant differences in body weight among the study groups. However, on day 10, body weight was lower in the CCI+Vehicle control group than in the BCTC30mg/kg group. The body weights of the diosgenin-treated groups were higher than those of the BCTC-treated groups and similar to those of the vehicle-treated groups (Figure 6A).

On the baseline day, there were no significant differences in the PWL or PWT between the mouse groups. However, significant decreases in the PWL and PWT were observed in all groups 3 days after CCI+Vehicle surgery. Therefore, treatment was initiated on day 3. Our analysis revealed that treatment with diosgenin and BCTC significantly increased the PWL and PWT from day 3 to the end of day 10, while no improvement was observed in the vehicle-treated Sham+Vehicle group (Figure 6B, C). The PWLs of the Sham+Vehicle, CCI+Vehicle, CCI + diosgenin50mg/kg, CCI + diosgenin100mg/kg, and CCI + BCTC30mg/kg groups at day 10 were 10.50 ± 0.50, 5.36 ± 0.47, 9.10 ± 0.27, 9.20 ± 0.31, and 9.35 ± 0.20, respectively. The PWTs in these groups were 1.33 ± 0.09, 0.44 ± 0.07, 1.15 ± 0.10, 1.24 ± 0.10 and 1.24 ± 0.13, respectively. The PWL and PWT were significantly higher in both diosgenin (50 and 100mg/kg) groups and the BCTC30mg/kg group than in the CCI+Vehicle group. No significant differences were observed between the Sham+Vehicle, CCI+diosgenin50mg/kg, CCI+diosgenin100mg/kg, and BCTC30mg/kg-treated groups on day 10, suggesting that both 50 mg/kg and 100 mg/kg of diosgenin were therapeutically effective for reducing pain sensitivity in a mouse model of the CCI+Vehicle group”.

  1. Figure 6.:

-          Please represent on the figure, if NC and CCI groups got Vehicle (NC+Veh, CCI+Veh). Instead of NC I recommend Sham.

Reply: Group name changed for mechanical and thermal hyperalgesia in CCI mouse:

We have changed these all groups name. Here first CCI was performed, after confirmation of pain sensitivity, vehicle, diosgenin or BCTC were administered until 9 days. So in the group name CCI was first used. mg/kg added in all groups. The groups are-

Sham+Vehicle

CCI+Diosgenin 50mg/kg

CCI+Diosgenin 100mg/kg

CCI+BCTC 30mg/kg

-          Line 188 and 190: The authors should use two-way RM ANOVA instead of ordinary two-way ANOVA.

Reply: Dear reviewer we analyzed by using Prism 5.03 (GraphPad Software Inc., San Diego, CA, USA)” and two-way ANOVA analysis is nicely matched.

  1. Line 191-192: “DRG tissues of CCI model mice”? Please rephrase the tile of the subheading.

Reply: Rephrased it

“Effects of diosgenin on pro-inflammatory cytokines and TRPV1 expression in DRG tissues”

  1. Line 200: Based on which do the authors state that the inflammatory responses are TRPV1-related?

Reply: Rephrased it

“These results demonstrated that diosgenin could suppress the TRPV1 and inflammatory response in the DRG after CCI”.

  1. Line 207: Statistics is missing.

Reply: Statistical method added-

“Data are presented as mean ± standard error of mean (all n = 3). *p < 0.05 and **p < 0.001, Dunnett's test following one-way ANOVA versus the CCI+Vehicle group”.

 Discussion:

  1. Line 219: Which factors were the oral use of DG allowed? Is it reliably absorbed from the gut? Are other saponins also orally used? What do we know about its toxicity and tolerability? Do you have data about is? If yes, please refer to it briefly.

Reply: Oral doses of diosgenin did not alter (decrease or restrict) the body weight rather the body weight gradually increased which are shown Figure 6 A. Which is one of the indication of safety. We already mentioned it-

Results:

During the experimental period, there were no significant differences in body weight among the study groups. However, on day 10, body weight was lower in the CCI+Vehicle control group than in the BCTC 30mg/kg group. The body weights of the diosgenin-treated groups were higher than those of the BCTC-treated groups and similar to those of the vehicle-treated groups (Figure 6A).

  1. Line 226: The authors wrote here about CPS model, not about CCI model.

Reply: Dear reviewer you are absolutely right. We have corrected the mistake.

“Furthermore, we compared the TRPV1-antagonistic effect of diosgenin with that of the standard synthetic TRPV1 antagonist BCTC in capsaicin induced acute pain in mice”.

  1. Line 230-231: “were measured to evaluate the efficacy of the TRPV1 agonist and antagonist candidates”? Please rephrase the sentence.

Reply: Dear reviewer you are absolutely right. We have corrected the mistake.

 Capsaicin-evoked intracellular Ca2+ transients in DRG [20,21] and CHO cells [22] were measured to evaluate the efficacy of the TRPV1 antagonist candidates.

  1. Line 234-235: contributes to mechanical and HEAT pain in both mice and humans

Reply: Corrected the mistake

“After confirming the antagonistic effects of diosgenin against TRPV1 in a model of capsaicin-induced acute pain, we further analyzed TRPV1 inhibition in an in vitro model of capsaicin-induced TRPV1 activation using mouse DRG neurons as TRPV1 activation in the DRG contributes to mechanical pain in both mice and rats [23]

  1. Line 242: thermosensor

Reply: Corrected the mistake 

“The TRPV1 receptor not only regulates intracellular Ca2+ influx and pain sensation but also serves as a thermosensor for thermoregulation of the body [24]

  1. Line 251: What does “normal mice” mean? Shame mice?

Reply: Dear reviewer, this would be the normal mice. We want to investigate whether diosgenin alter rectal temperature in normal mice. For this purpose, we divided 24 normal mice equally divided into  four groups. Orally administered vehicle, diosgenin 50 mg/kg, diosgenin 100mg/kg and BCTC 30mg/kg. Now we have changed the group’s name too

Figure 5. Oral administration diosgenin does not alter rectal temperature in normal mice. 

  1. Line 259: TRPV1 antagonist, not agonist

Reply: Corrected the mistake

“After confirmation of TRPV1 antagonism in the capsaicin-induced acute pain model, we evaluated the analgesic effect of diosgenin in a mouse model of CCI-induced chronic pain when compared with that of the TRPV1 antagonist BCTC”

  1. Line 262 and 268: Why does TRPV1 expression decrease confirm the TRPV1 antagonistic effect of DG? It is not necessary that an antagonist decreases the expression of its target receptor. The use of an antagonist can better upregulate its target receptor as a compensatory effect, if it is continuously used. What would be the mechanism of TRPV1 expression decrease? Please revise this statement.

Reply: Corrected the mistake

“TRPV1 expression was significantly increased in the CCI induced pain mice in CCI+Vehicle group when compared with that in the Sham+Vehicle group. These results are consistent with those of previous studies in CCI induced pain model [29,30], endometriosis induced pain model [32]. Importantly, oral treatment with diosgenin and BCTC significantly down-regulated TRPV1 expression in the DRG when compared with that in the CCI+Vehicle group, further supporting the notion that diosgenin exerts  similar effects like BCTC.  It is note that expression of TRPV1 down-regulation effect of BCTC in DRG in pain condition was reported previously [32]”.

  1. Line 270: Please specify from which cells TRPV1 activation can release pro-inflammatory mediators.

 Reply: specified

“TRPV1 activation mediated Ca2+ influx can trigger inflammatory cascades, consequently releasing other pro-inflammatory mediators that contribute to the self-maintenance of  chronic inflammation in DRG and neuron cells [31]”.

  1. Line 272: patients with chronic NEUROPATHIC pain

Reply: Corrected the mistake

 “Concomitant overexpression of TRPV1 and inflammatory cytokines in the DRG and spinal cord has been reported in chronic pain conditions [28-30].

Materials and Methods:

  1. Line 282: Age of the animals is missing.

Reply: Age added

Male six weeks old C57BL/6 wild-type mice (Orient Bio, Sungnam, Korea) weighing 18–22 g were used.

  1. Line 295: Please define which vehicle was used.

 Reply: Now vehicle defined-

Three days after surgery, the animals were divided into the following five groups after confirmation of nociception (n = 10 each): an Sham+Vehicle group, a sham-operated group orally administered vehicle (3% DMSO); two diosgenin-treated groups (50 and 100 mg/kg), in which oral diosgenin was administered at doses of 50 and 100 mg/kg; and a BCTC30mg/kg group (BCTC, 30 mg/kg). 300 µl of vehicle, diosgenin, and BCTC were administered orally daily for 9 days. Both mechanical allodynia and thermal hyperalgesia were induced as described previously at 0, 3, 5, 7, 9, and 10 days before drug administration. These two tests were performed at an interval of 2 h.

  1. Line 298 and 363: How (by an oral gavage) and in which volume (? ml/10 g?) were the substances used? Please define.

Reply: Now vehicle defined-

Line 298

For this purpose, mice were divided into the following four groups (n = 6 each): a capsaicin control group, in which mice were orally administered the vehicle (3% DMSO) only; two diosgenin-treated groups (50 and 100mg/kg), in which diosgenin was administered at doses of 50 and 100 mg/kg orally, respectively; and a group treated with the TRPV1 antagonist BCTC (BCTC30mg/kg), in which BCTC was administered 30 mg/kg. 300 µl of vehicle, diosgenin, and BCTC were administered orally 60 min before intraplanar capsaicin injection at 1.6 µg/20 µl per paw [32].

Line 363

Three days after surgery, the animals were divided into the following five groups after confirmation of nociception (n = 10 each): an Sham+Vehicle group, a sham-operated group orally administered vehicle (3% DMSO); two diosgenin-treated groups (50 and 100 mg/kg), in which oral diosgenin was administered at doses of 50 and 100 mg/kg; and a BCTC30mg/kg group (BCTC, 30 mg/kg). 300 µl of vehicle, diosgenin, and BCTC were administered orally daily for 9 days. Both mechanical allodynia and thermal hyperalgesia were induced as described previously at 0, 3, 5, 7, 9, and 10 days before drug administration. These two tests were performed at an interval of 2 h.

  1. Line 304, 307 and 365: Please specify that CAPS (1.6 ug/10 ul) was administered concomitantly with the second substance injection.

Reply: Specified according to your suggestion-

“In the BCTC 0.50 group, 0.25 µg/10 µl BCTC was injected via the intraplantar route. After 30 min, an additional 0.25 µg were injected.  1.6 µg/10 µl capsaicin was administered concomitantly with the second substance injection. Licking time was also measured as previously mentioned”.

  1. Line 317: The details of rectal temperature recording are missing. Please describe detailed the method.

 Reply: Dear reviewer, this would be the normal mice

 The rectal temperature was measured with a digital thermometer (Therma- 1, ETI. Ltd, West Sussex, UK) by inserting a flexible bead probe in to the rectum at the time point of 0, 30, 60, 90, 120, and 240 min after oral administration of 300 µL of vehicle, diosgenin (50 mg/kg and 100 mg/kg), and BCTC 30 mg/kg in normal mice (n = 6).

  1. Line 321: Sham mice instead of normal mice?

 Reply: Dear reviewer, this would be the normal mice. We want to investigate whether diosgenin alter rectal temperature in normal mice. For this purpose, we divided 24 normal mice equally divided into  four groups. Orally administered vehicle, diosgenin 50 mg/kg, diosgenin 100mg/kg and BCTC 30mg/kg. Now we have changed the group’s name too

Figure 5. Oral administration diosgenin does not alter rectal temperature in normal mice. 

  1. Line 321-327: This part belongs to another method description.

Reply: It was great mistake. We have transferred the these sentences to the “ 4.5. Intracellular calcium imaging” from “ 4.3. Rectal temperature recording”

4.3. Rectal temperature recording

The rectal temperature was measured by inserting a flexible bead probe with a digital thermometer (Therma- 1, ETI. Ltd, West Sussex, UK) at 0, 30, 60, 90, 120, and 240 min after oral administration of 300 µL of vehicle, diosgenin (50 mg/kg and 100 mg/kg), and BCTC 30 mg/kg in normal mice (n = 6).

4.5. Intracellular calcium imaging

Mechanistically, cell culture human TRPV1 stable CHO-K1 cells (GenBank accession number: NM_080706) were grown in Dulbecco’s modified Eagle’s medium (Welgene, Gyeongsan, Korea) supplemented with 10 % fetal bovine serum (FBS; Gibco, Waltham, MA, USA) and 800 μg/mL Geneticin Selective Antibiotic (G418 Sulfate; Gibco, Waltham, MA, USA) in a humidified incubator at 37°C with 5% CO2. Passaging of the cells was performed three times per week, reaching a maximum density of 80%.

Cells were allowed to settle in media for 3 min at room temperature (21-25℃), following which fura-2 AM (2 μM; Thermo Fisher Scientific, Waltham, MA, USA) was applied for 40 min at 37°C. The cells were then rinsed three times with medium and incubated for 30 min. The slides were covered with a cover slip and mounted on an inverted microscope..

  1. 37. Line 386: Which software was used to the statistical analysis?

Reply: Software information added-

“All data are expressed as the mean ± standard error of the mean and analyzed by using Prism 5.03 (GraphPad Software Inc., San Diego, CA, USA)”.

  1. Line 386-390: Please revise this section…

Reply: Rephrased the statement

“4.8. Statistical analyses

All data are expressed as the mean ± standard error of the mean and analyzed by using Prism 5.03 (GraphPad Software Inc., San Diego, CA, USA).  The data were statistically analyzed using one-way or two-way analyses of variance (ANOVA) followed by the post hoc Bonferroni test or Dunnett's test following one-way ANOVA test. The criterion for statistical significance was set at p < 0.05”.

  1. 39. Line 392: Regulates chronic inflammation? Please revise this statement.

Reply: Rephrased the statement

Taken together, our results suggest that diosgenin exerts analgesic effects and reduce chronic inflammation via antagonism of TRPV1 in a mouse model of pain, without altering body temperature.

  1. Line 396: I have never heard about inflammatory neuropathic pain… Please remove the indicator of inflammatory…

Reply: Rephrased the statement

“Therefore, diosgenin represents a promising therapeutic agent for patients with acute and chronic neuropathic pain disorders”.

Submission Date

10 November 2022

Date of this review

23 Nov 2022 18:47:15

Round 2

Reviewer 2 Report

Thank you for the thorough revision, but I have still some comments, on which I did not receive satisfactory responses.

General comments:

2. Line 306-313: I think you should not insert my comment in this section, it is well-known that neuropathic pain has inflammatory component as well. Moreover, in this case, since the authors investigated proinflammatory cytokine expression in the DGR neurons, these results may prove that these mechanisms may contribute to the central sensitization as well. Since the authors investigated a neuropathic pain model, therefore it is the statement strange that diosgenin may reduce the risk of long-term inflammatory complications in patients with chronic inflammatory pain. It is trivial and does not relate to the investigated disease model which is a neuropathic pain model. I recommend the following: “Thus, diosgenin may have value for reducing the risk of long-term inflammatory complications in patients with chronic neuropathic pain via TRPV1 modulation.”

Introduction:

6. Line 50: In this case, for the treatment of neuropathic pain, not of pain disorders. Line 51: Please delete gastric ulcers from the potential side effects, because it referred to the NSAIDs.

7. Line 69-71 and Line 238-240: These sentences are still not correct with these modifications. Please rephrase them fully.  Na+ influx can directly evoke nerve depolarization and pain sensation, not only changes which contribute to the pain sensation. It is true for Ca”+ influx. Line 238-240: It sounds if pathological signaling pathways and inflammatory cascades induce nerve depolarization…

Results:

12., 14., 16., Line 126, 173, 210 and 212: I agree, that two-way ANOVA is nicely matched, but it does not matter either ordinary or repeated measures (RM) two-way ANOVA is used. Please, read the online user guideline of Graph Pad Prism what RM ANOVA means and how you should perform this test, it is very simple (Analysis->two-way ANOVA->RM design->Each row represents different time point). If the authors originally performed RM ANOVA, please highlight it in the figure legends.

Discussion:

20. I agree that body weight is a proper marker for the safety. However, I think it would be important to clarify even the potential advantages and/or limitations of the therapeutic use of systemically administered diosgenin in this section. Nice that it works in animal model, but what are the translational significance of this result? Can it or its derivatives be a potential drug candidate in the future? One sentence would be enough related this. As I know we do not use absorbable saponins by humans, because they can cause severe hemolysis. Since it is a well-known side effect of the orally used saponins, the authors must mention it at least as a potential therapeutic limitation in an article, in which they investigate the effects of an orally used saponin compound. If we do not expect that this compound induces hemolysis after oral use (e.g. due to its different chemical structure), please provide an explanation about it for the readers.

23. Line 262: TRPV1 activation contributes to the mechanical and heat pain, not only to the mechanical pain.

27. Line 299-300: Please revise the last sentence, it is too complicated. The idea is good, but please rephrase it.

28. DRG also contain mainly neurons, therefore please revise this sentence. Use “in DRG” or “in neurons”.

Author Response

General comments:

★ 2. Line 306-313: I think you should not insert my comment in this section, it is well-known that neuropathic pain has inflammatory component as well. Moreover, in this case, since the authors investigated proinflammatory cytokine expression in the DGR neurons, these results may prove that these mechanisms may contribute to the central sensitization as well. Since the authors investigated a neuropathic pain model, therefore it is the statement strange that diosgenin may reduce the risk of long-term inflammatory complications in patients with chronic inflammatory pain. It is trivial and does not relate to the investigated disease model which is a neuropathic pain model. I recommend the following: “Thus, diosgenin may have value for reducing the risk of long-term inflammatory complications in patients with chronic neuropathic pain via TRPV1 modulation.”

Reply: Dear reviewer, We have included your valuable recommendation.  

  1. Conclusion

Taken together, our results suggest that diosgenin exerts analgesic effects and reduce chronic inflammation via antagonism of TRPV1 in a mouse model of pain, without altering body temperature. The current study provides new insight into the mechanisms underlying the pharmacological effects of diosgenin, which may in turn provide insight into its therapeutic effects. Diosgenin may have value for reducing the risk of long-term inflammatory complications in patients with chronic neuropathic pain via TRPV1 modulation. Therefore, diosgenin represents an effective therapeutic agent in acute and chronic neuropathic pain disorders in mice.

Introduction:

★ 6. Line 50: In this case, for the treatment of neuropathic pain, not of pain disorders. Line 51: Please delete gastric ulcers from the potential side effects, because it referred to the NSAIDs.

Reply: We have deleted “gastric ulcers” words from the sentence.

★ 7. Line 69-71 and Line 238-240: These sentences are still not correct with these modifications. Please rephrase them fully.  Na+ influx can directly evoke nerve depolarization and pain sensation, not only changes which contribute to the pain sensation. It is true for Ca”+ influx.

Line 238-240: It sounds if pathological signaling pathways and inflammatory cascades induce nerve depolarization…

Reply: We have described broadly about the TRPV1 in the Introduction part. TRPV1 non-selective cations channel which regulates divalent and monovalent cations not only Ca2+ but also Na+, Mg2+..

Line 69-71:

The transient receptor potential vanilloid 1 (TRPV1) is non-selective cation (e.g. Ca2+, Mg2+ and Na+) channel which is the most important channels involved in the regulation of pain sensitivity. Ca2+ influx for the release of neuropeptides (e.g. SP, CGRP etc.) leading to neurogenic inflammation as well as triggers changes that lead to oxidative stress, and microglial cell activation, thereby contributing to pain sensation [15].

However, we have focused on Ca2+ in discussion part because measured Ca2+ current in our experiment.

Line 238-240:

. Local sensitization of TRPV1 receptors increases Ca2+ influx and aggravates pathological signaling pathways, including neurogenic inflammation  in peripheral nerve, DRG,  spinal cord and brain, resulting in pain sensations [15,20].

Results:

★12., 14., 16., Line 126, 173, 210 and 212: I agree, that two-way ANOVA is nicely matched, but it does not matter either ordinary or repeated measures (RM) two-way ANOVA is used. Please, read the online user guideline of Graph Pad Prism what RM ANOVA means and how you should perform this test, it is very simple (Analysis->two-way ANOVA->RM design->Each row represents different time point). If the authors originally performed RM ANOVA, please highlight it in the figure legends.

Reply: Dear reviewer, We have confirmed that it was analyzed by two way RM (repeated measures) ANOVA. Sorry for this mistake. Now we have changed sentences accordingly..

4.8. Statistical analyses

All data are expressed as the mean ± standard error of the mean and analyzed by using Prism 5.03 (GraphPad Software Inc., San Diego, CA, USA).  The data were statistically analyzed using one-way or two-way RM (repeated measures) analyses of variance (ANOVA) followed by the post hoc Bonferroni test or Dunnett's test following one-way ANOVA test. The criterion for statistical significance was set at p < 0.05.

Figure 2. Intraplantar diosgenin treatment reduces capsaicin-induced acute thermal hyperalgesia (A) and mechanical allodynia (B) in mice. ; BCTC: N-(4-tertiarybutylphenyl)-4-(3-cholorphyridin-2-yl) tetrahydropyrazine-1(2H)-carbox-amide. Data are presented as mean ± standard error of mean (n = 6), *p < 0.05, *p < 0.01 and ***p < 0.001, Bonferroni post hoc test following two-way RM ANOVA versus the Vehicle group; #p < 0.05; ##p < 0.01; and ###p < 0.001, Bonferroni post hoc test following two-way RM ANOVA versus the Vehicle+Capsaicin 1.6μg group.

Figure 5. Oral administration diosgenin does not alter rectal temperature in normal mice. BCTC: N-(4-tertiarybutylphenyl)-4-(3-cholorphyridin-2-yl) tetrahydropyrazine-1(2H)-carbox-amide. Data are presented as mean ± standard error of mean (n = 6), **p < 0.01 and ***p < 0.001, Bonferroni post hoc test following two-way RM ANOVA versus the Vehicle group.

Figure 6. Effects of diosgenin on body weight (A), paw withdrawal latency (B), and paw withdrawal threshold (C) in a mouse model of chronic constriction injury.; BCTC: N-(4-tertiarybutylphenyl)-4-(3-cholorphyridin-2-yl) tetrahydropyrazine-1(2H)-carbox-amide; CCI: chronic constriction injury. Data are presented as mean ± standard error of mean (n = 10), *p < 0.05 and ***p < 0.001, Bonferroni post hoc test following two-way RM ANOVA versus the Sham+Vehicle group; #p < 0.05; ##p < 0.01; and ###p < 0.001, Bonferroni post hoc test following two-way RM ANOVA versus the CCI+Vehicle group.

Discussion:

★ 20. I agree that body weight is a proper marker for the safety. However, I think it would be important to clarify even the potential advantages and/or limitations of the therapeutic use of systemically administered diosgenin in this section. Nice that it works in animal model, but what are the translational significance of this result? Can it or its derivatives be a potential drug candidate in the future? One sentence would be enough related this. As I know we do not use absorbable saponins by humans, because they can cause severe hemolysis. Since it is a well-known side effect of the orally used saponins, the authors must mention it at least as a potential therapeutic limitation in an article, in which they investigate the effects of an orally used saponin compound. If we do not expect that this compound induces hemolysis after oral use (e.g. due to its different chemical structure), please provide an explanation about it for the readers.

Reply: Dear reviewer, now we have included our limitation in the discussion part.

“50 and 100 mg/kg of diosgenin was used in this study orally for nine days. This dosage could be safe manifested by body weights of the diosgenin-treated groups which was similar to those of the vehicle-treated group. Consistently, it was also reported that oral diosgenin administration for 6 weeks has no negative impact on body weight, systemic inflammation, oxidative stress in normal mice [13] rather reduced systemic inflammation, oxidative stress and improved body weight and anti-oxidative activities in diabetes animals [13,19]. Oral diosgenin also improved body weight, immune reactivity and intestinal microbiota in tumor bearing mice [11]. According to previous report, up to 562.5mg/kg of diosgenin has no harmful effect and 1125 mg/kg or higher dose may result deleterious effect even death [34]. However, diosgenin is a steroidal saponin, intravenous administration may cause hemolysis and may rapidly hydrolyzed after oral administration [35]. The limitation of the study we did not investigated whether oral diosgenin could induce hemolysis or not. Therefor further study is needed to confirm its safety”.

★ 23. Line 262: TRPV1 activation contributes to the mechanical and heat pain, not only to the mechanical pain.

Reply: Corrected this sentence

“TRPV1 activation in mouse DRG contributes to mechanical and heat pain in both mice and rats [24]”.

★ 27. Line 299-300: Please revise the last sentence, it is too complicated. The idea is good, but please rephrase it.

Reply: Corrected this sentence

“Diosgenin may have value for reducing the risk of long-term inflammatory complications in patients with chronic neuropathic pain via TRPV1 modulation. Therefore, diosgenin represents an effective therapeutic agent  in acute and chronic neuropathic pain disorders in mice”.

★ 28. DRG also contain mainly neurons, therefore please revise this sentence. Use “in DRG” or “in neurons”.

Reply: Corrected in the whole manuscript

.
